# Relationship between body mass index and clinical events in patients with atrial fibrillation undergoing percutaneous coronary intervention

Tatsuro Yamazaki[1], Hideki Kitahara[1]*, Daichi Yamashita[1], Takanori Sato[1], Sakuramaru Suzuki[2], Takashi Hiraga[1], Tadahiro Matsumoto[1], Takahiro Kobayashi[1], Yuji Ohno[3], Junya Harada[4], Kenichi Fukushima[5], Tatsuhiko Asano[6], Naoki Ishio[7], Raita Uchiyama[8], Hirofumi Miyahara[9], Shinichi Okino[10], Masanori Sano[11], Nehiro Kuriyama[12], Masashi Yamamoto[13], Naoya Sakamoto[14], Junji Kanda[15], Yoshio Kobayashi[1]

1 Department of Cardiovascular Medicine, Chiba University Graduate School of Medicine, Chiba, Japan, 2 Department of Cardiovascular Medicine, Eastern Chiba Medical Center, Togane, Japan, 3 Department of Cardiovascular Medicine, Narita Red Cross Hospital, Narita, Japan, 4 Division of Cardiology, Chiba Cerebral and Cardiovascular Center, Ichihara, Japan, 5 Department of Cardiology, Matsudo City General Hospital, Matsudo, Japan, 6 Department of Cardiology, Chiba Rosai Hospital, Ichihara, Japan, 7 Department of Cardiology, Chiba Aoba Municipal Hospital, Chiba, Japan, 8 Department of Cardiovascular Medicine, Japan Community Healthcare Organization Chiba Hospital, Chiba, Japan, 9 Department of Cardiology, Chiba Kaihin Municipal Hospital, Chiba, Japan, 10 Department of Cardiology, Funabashi Municipal Medical Center, Funabashi, Japan, 11 Department of Cardiology, Chiba Emergency Medical Center, Chiba, Japan, 12 Cardiovascular Center, Miyazaki Medical Association Hospital, Miyazaki, Japan, 13 Department of Cardiology, Kimitsu Central Hospital, Kisarazu, Japan, 14 Division of Cardiology, Chibaken Saiseikai Narashino Hospital, Narashino, Japan, 15 Department of Cardiovascular Medicine, Asahi General Hospital, Asahi, Japan

* hidekitahara0306@gmail.com

**Data Availability Statement:** All relevant data are within the manuscript and its Supporting Information files.

## Abstract

### Background

It is still unclear whether body mass index (BMI) affects bleeding and cardiovascular events in patients requiring oral anticoagulants (OAC) for atrial fibrillation (AF) and antiplatelet agents after percutaneous coronary intervention (PCI) for coronary artery disease (CAD). The aim of this study was to evaluate the relationship between BMI and clinical events in patients who underwent PCI under OAC therapy for AF.

### Method

This was a multicenter, observational cohort study conducted at 15 institutions in Japan. AF patients who underwent PCI with drug-eluting stents for CAD were retrospectively and prospectively included. Patients were divided into the Group 1 (BMI <21.3 kg/m²) and the Group 2 (BMI ≥21.3 kg/m²) according to the first-quartile value of BMI. The primary endpoint was net adverse clinical events (NACE), a composite of major adverse cardiovascular events (MACE) and major bleeding events within one year after index PCI procedure.

**Funding:** The author(s) received no specific funding for this work.

**Competing interests:** The authors have declared that no competing interests exist.

## Results

In the 720 patients, 180 patients (25.0%) had BMI value <21.3 kg/m$^2$. While the rates of NACE and MACE were significantly higher in the Group 1 than the counterpart (21.1% vs. 11.9%, p = 0.003 and 17.2% vs. 8.9%, p = 0.004), that of major bleeding did not differ significantly between the 2 groups (5.6% vs. 4.3%, p = 0.54). The cumulative rate of NACE and MACE was significantly higher in the Group 1 than the Group 2 (both log-rank p = 0.002), although that of major bleeding events was equivalent between the 2 groups (log-rank p = 0.41). In multivariable Cox regression analyses, while BMI value <21.3 kg/m$^2$ was not associated with major bleeding events, that cut-off value was an independent predictor for increased NACE and MACE.

## Conclusions

Among the patients undergoing PCI for CAD and requiring OAC for AF, BMI value was a useful indicator to predict major adverse clinical events.

## Introduction

Body mass index (BMI), calculated as body weight in kilograms divided by height in meters squared, is a common and simple indicator, which can evaluate physique easily in daily clinical practice. In general, patients with a high value of BMI are likely to have chronic diseases associated with progression of arteriosclerosis, such as diabetes, hypertension, and dyslipidemia [1, 2], resulting in increased cardiovascular events and mortality [1, 3]. On the other hand, several reports have suggested that patients with low BMI values have a higher incidence of mortality and cardiovascular events than those with normal and high range of BMI in populations with cardiovascular disease, including coronary artery disease, heart failure, and atrial fibrillation (AF) [4–7]. These phenomena have been proposed as "obesity paradox" and still remain controversial. Given that BMI value is associated with the prognosis of patients, it is clinically important to include optimal BMI and body weight for preventing cardiovascular events.

AF is a common cardiac arrhythmia worldwide, and its prevalence is reportedly associated with BMI value [8]. Due to increasing average life expectancy, improving survival rate of various diseases, and facilitation of rhythm monitoring devices [8, 9], the number of patients with AF is increasing, accompanied by the increasing number of patients with AF undergoing percutaneous coronary intervention (PCI) for concomitant coronary artery disease [10]. Previous studies have reported that approximately 10% of patients have a history of AF when they undergo PCI [11, 12]. While using oral anticoagulant (OAC), including vitamin K antagonist (VKA) or direct oral anticoagulant (DOAC), it is recommended as standard therapy for AF patients to prevent systemic embolism [13–15]. Moreover, OAC is recognized as an important risk factor for bleeding complications in patients who need antiplatelet drugs after PCI [16, 17]. The recent Japanese guideline pertaining to antithrombotic therapy for patients with coronary artery disease (CAD) demonstrated that both low body weight and OAC are important risk factors for major bleeding complications [16]. Given that Asians, especially East Asians, are likely to have lower body weight compared with non-Asians [18, 19], it is clinically relevant to evaluate the risk of bleeding events as well as adverse cardiovascular events in patients with low BMI requiring OAC for AF and antiplatelet drugs after PCI. Thus, the aim of this study

was to evaluate the relationship between BMI and clinical events in patients who underwent PCI under OAC therapy for AF.

## Materials and methods

### Study population

This was a multicenter and observational cohort study performed in 15 hospitals in Japan (CHIBA AF-PCI registry) [20]. The study flow is summarized in Fig 1. We enrolled the AF patients who underwent PCI with drug-eluting stents (DES) for coronary artery disease prospectively from 9th July 2019 to 26th March 2021 and included them retrospectively from 1st June 2015 to 31st March 2021 (n = 949). The main exclusion criteria were as follows: PCI with two stent technique for bifurcation lesion, cardiogenic shock, history of stent thrombosis, history of aortic or mitral valve replacement, and severe liver dysfunction. Next, patients with OAC used for reasons other than AF (n = 11), no DES implantation (n = 25), no PCI information (n = 3), no OAC used (n = 134), and no detailed follow-up information (n = 30) were excluded. Last, patients who discontinued OAC without any clinical events (n = 17) and those without BMI data (n = 9) were also excluded. Thus, 720 patients were included in the present analysis, and were divided into 2 groups according to their BMI value. Patients who had less than the first quartile value of BMI were classified as "Group 1" and remaining patients were classified as "Group 2" (Fig 1). This study was performed in accordance with the principles of the Declaration of Helsinki and registered in the University Hospital Medical Information Network (UMIN) Clinical Trials Registry (ID: UMIN000047503). The study protocol was approved by the ethics committee of Chiba University Graduate School of Medicine (unique identifier: 3443), and also approved by the institutional review board or ethics committee at each participating institution. Prospectively enrolled patients provided written informed consent for research before study entry. In retrospectively included patients, informed consent was ascertained in the form of opt-out. An anonymization process was performed, removing all identifiable information. Authors did not access to information that could identify

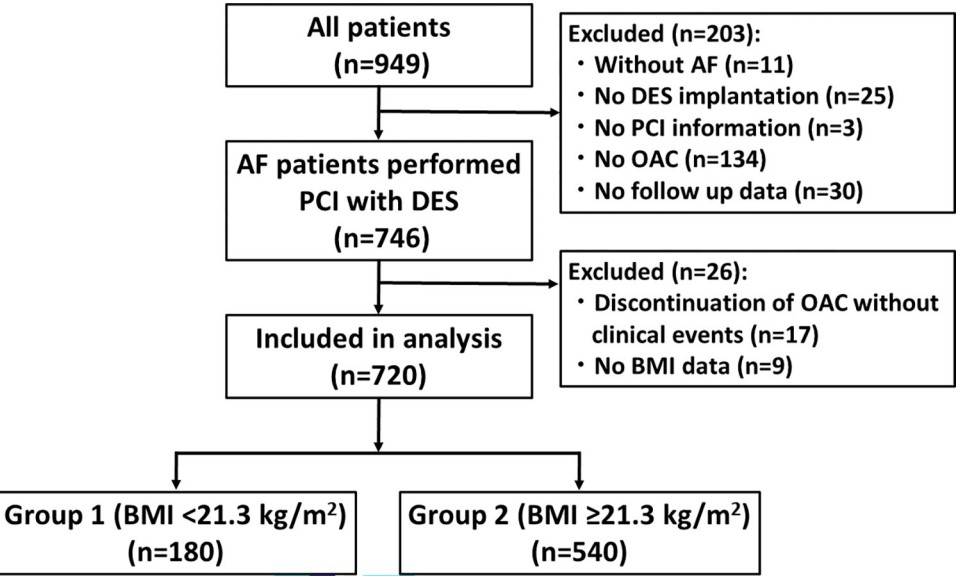

**Fig 1. Study flow.** AF, atrial fibrillation; BMI, body mass index; DES, drug-eluting stents; OAC, oral anticoagulant; PCI, percutaneous coronary intervention.

## Medical treatment

In this study, PCI was performed per local protocol using DES in each institution [21, 22]. Antithrombotic therapy for patients requiring OAC after performing PCI was left to the operator's discretion based on recent recommendations and guidelines [16, 17, 23]. In Japan, prasugrel has been used with OAC due to the approval of low-dose prasugrel (loading/maintenance dose: 20/3.75 mg/day), which is about one-third of that in other countries, as an antithrombotic therapy after performing PCI [24, 25]. In addition, optimal medical therapy and life guidance to prevent recurrence of cardiovascular disease were left to operators in each institution.

## Clinical events

In this study, the primary endpoint was the cumulative incidence of net adverse clinical events (NACE), a composite of major adverse cardiovascular events (MACE) and major bleeding events within 1 year after index PCI procedure [26, 27]. MACE was defined as a composite of all-cause death, non-fatal myocardial infarction, stent thrombosis, and stroke. Major bleeding events were defined as Bleeding Academic Research Consortium (BARC) types 3 or 5. The secondary endpoints were the cumulative incidence MACE and major bleeding events, respectively.

## Statistical analysis

The main purpose of this study was to compare clinical outcomes between the Group 1 and 2. The categorical variables were shown as n (%) and analyzed with Fisher's exact test or Pearson's chi-square test. The continuous variables were presented as mean ± standard deviation and analyzed with one-way analysis of variance (ANOVA). To evaluate the cumulative events of NACE, MACE and major bleeding, Kaplan-Meier curve analysis and log-lank test were performed. Multivariable Cox regression analysis was performed to calculate the hazard ratio (HR) to NACE, MACE, and major bleeding events. Variables included in the multivariable analysis were selected referencing the recent Japan circulation society guideline and previous reports [16, 28], in which the factors associated with thrombotic events were included in the analysis of MACE and those proposed as predictors for bleeding events in the Japanese high bleeding risk were done in analysis of major bleeding. In addition, the multivariable analysis of NACE included the variables associated with thrombotic and major bleeding events. As a subanalysis, propensity score matching was performed with propensity score calculated using logistic regression analysis to adjust between-group differences in age. In this study, the patients with eGFR $<30$ml/min/1.73 m$^2$, platelets level $<10\times10^4$/μL, or hemoglobin level $<11$g/dl were defined as having severe chronic kidney dysfunction (CKD), thrombocytopenia, or moderate to severe anemia, respectively [16]. A value of p $<0.05$ was considered statistically significant. All statistical analyses were performed with JMP pro version 16.0 (SAS Institute Inc., Cary, USA) and EZR version 1.55 (Saitama Medical Center, Jichi Medical University, Saitama, Japan), which is a graphical user interface for R version 2.7–1 (The R Foundation for Statistical Computing, Vienna, Austria).

## Results

In a total of 720 patients enrolled, the first quartile value of BMI was 21.3 kg/m$^2$, and based on this value, patients were divided into the Group 1 (n = 180) and Group 2 (n = 540) groups. Baseline characteristics are shown in Table 1.

**Table 1. Baseline characteristics.**

| Variables | All (n = 720) | Group 1 (n = 180) | Group 2 (n = 540) | p value |
|---|---|---|---|---|
| Age (years) | 74.0±9.0 | 77.3±8.4 | 72.8±9.0 | <0.001 |
| Male | 580 (80.6%) | 132 (73.3%) | 448 (83.0%) | 0.006 |
| Body weight (kg) | 64.8±14.1 | 50.5±7.4 | 69.6±12.6 | <0.001 |
| BMI (kg/m$^2$) | 24.4±4.3 | 19.5±1.5 | 26.0±3.6 | <0.001 |
| Hypertension | 610 (84.7%) | 145 (80.6%) | 465 (86.1%) | 0.09 |
| Diabetes | 352 (48.9%) | 75 (41.7%) | 277 (51.3%) | 0.03 |
| Dyslipidemia | 535 (74.3%) | 119 (66.1%) | 416 (77.0%) | 0.004 |
| Current smoking | 131 (18.3%) | 22 (12.2%) | 109 (20.4%) | 0.01 |
| Family history of CAD | 95 (14.5%) | 17 (10.2%) | 78 (16.0%) | 0.07 |
| Prior angina pectoris | 203 (28.2%) | 45 (25.0%) | 158 (29.3%) | 0.29 |
| Previous MI | 186 (25.8%) | 45 (25.0%) | 141 (26.1%) | 0.84 |
| Prior PCI | 238 (33.1%) | 51 (28.3%) | 187 (34.6%) | 0.14 |
| Prior CABG | 37 (5.1%) | 14 (7.8%) | 23 (4.3%) | 0.08 |
| Hemodialysis | 20 (2.8%) | 7 (3.9%) | 13 (2.4%) | 0.30 |
| Paroxysmal AF | 375 (52.2%) | 93 (51.7%) | 282 (52.3%) | 0.93 |
| Peripheral artery disease | 55 (7.6%) | 13 (7.2%) | 42 (7.8%) | 0.87 |
| Prior heart failure | 185 (25.7%) | 59 (32.8%) | 126 (23.4%) | 0.01 |
| Prior major bleeding events | 45 (6.3%) | 15 (8.3%) | 30 (5.6%) | 0.21 |
| Severe CKD | 71 (9.9%) | 17 (9.4%) | 54 (10.0%) | 0.89 |
| Thrombocytopenia | 12 (1.7%) | 5 (2.8%) | 7 (1.3%) | 0.19 |
| Moderate to severe anemia | 114 (15.8%) | 45 (25%) | 69 (12.8%) | <0.001 |
| Liver cirrhosis | 3 (0.4%) | 3 (1.7%) | 0 (0.0%) | 0.02 |
| Active cancer | 40 (5.6%) | 15 (8.3%) | 25 (4.6%) | 0.09 |
| Prior bleeding stroke | 22 (3.1%) | 5 (2.8%) | 17 (3.2%) | 1.00 |
| Prior ischemic stroke | 141 (19.9%) | 36 (20.0%) | 105 (19.4%) | 0.91 |
| LVEF (%) | 51.3±14.3 | 49.5±15.2 | 51.9±14.0 | 0.06 |
| CHADS2 score | 2.6±1.3 | 2.8±1.3 | 2.6±1.3 | 0.10 |
| CHA2DS2-VASc score | 3.8±1.6 | 4.0±1.5 | 3.7±1.6 | 0.01 |
| HAS-BLED score | 3.2±0.9 | 3.3±1.0 | 3.1±0.9 | 0.06 |
| Laboratory data | | | | |
| Hemoglobin (g/dl) | 13.1±2.2 | 12.3±2.2 | 13.4±2.1 | <0.001 |
| Platelet (×10$^4$/μL) | 20.4±6.4 | 21.7±10.6 | 20.2±6.1 | 0.03 |
| eGFR (ml/min/1.73 m$^2$) | 53.9±18.9 | 52.2±20.3 | 54.5±18.4 | 0.21 |
| Total cholesterol (mg/dL) | 167.6±38.7 | 167.9±39.7 | 167.5±38.4 | 0.89 |
| HDL cholesterol (mg/dL) | 48.8±14.1 | 52.5±15.4 | 47.5±13.5 | <0.001 |
| LDL cholesterol (mg/dl) | 96.1±31.9 | 94.7±33.0 | 96.6±31.6 | 0.50 |
| Triglycerides (mg/dL) | 129.6±92.7 | 95.7±51.4 | 141.1±100.4 | <0.001 |
| Glycated hemoglobin (%) | 6.4±1.1 | 6.3±1.2 | 6.4±1.0 | 0.43 |
| Lesion and procedure | | | | |
| Acute coronary syndrome | 288 (40.0%) | 84 (46.7%) | 204 (37.8%) | 0.04 |
| Multivessel disease | 329 (45.8%) | 74 (41.3%) | 255 (47.2%) | 0.19 |
| Bifurcation lesion | 133 (18.5%) | 37 (20.7%) | 96 (17.8%) | 0.44 |
| Femoral approach | 112 (15.8%) | 26 (14.6%) | 86 (16.1%) | 0.72 |
| Mechanical support | 23 (3.2%) | 12 (6.7%) | 11 (2.0%) | 0.005 |
| DES type | | | | |
| Everolimus | 444 (61.7%) | 111 (61.7%) | 333 (61.7%) | 0.20 |

(*Continued*)

**Table 1.** (Continued)

| Variables | All (n = 720) | Group 1 (n = 180) | Group 2 (n = 540) | p value |
|---|---|---|---|---|
| Zotarolimus | 100 (13.9%) | 28 (15.6%) | 72 (13.3%) | |
| Sirolimus | 121 (16.8%) | 30 (16.7%) | 91 (16.9%) | |
| Biolimus | 11 (1.5%) | 5 (2.8%) | 6 (1.1%) | |
| Multiple | 44 (6.1%) | 6 (3.3%) | 38 (7.0%) | |
| Number of stents | 1.5±0.8 | 1.5±0.8 | 1.5±0.8 | 0.35 |
| Mean stent diameter (mm) | 3.0±0.5 | 2.9±0.5 | 3.0±0.5 | 0.20 |
| Total stent length (mm) | 37.3±22.9 | 36.3±22.2 | 37.6±23.1 | 0.49 |

AF, atrial fibrillation; BMI, Body mass index; CABG, coronary artery bypass grafting; CAD, coronary artery disease; CKD, chronic kidney disease; DES, drug-eluting stent; eGFR, estimated glomerular filtration rate; HDL, high-density lipoprotein; LDL, low-density lipoprotein; LVEF, left ventricular ejection fraction; MI, myocardial infarction; PCI, percutaneous coronary intervention.

The mean BMI values were 19.5±1.5 and 26.0±3.6 kg/m$^2$, for the Group 1 and 2, respectively. Age was significantly higher, and the proportion of men was lower in the Group 1. The prevalence of conventional risk factors for cardiovascular disease, such as diabetes, dyslipidemia, and current smoking, were significantly higher in the Group 2. CHA2DS2-VASc and HAS-BLED scores were higher in the Group 1 than the counterpart. Patients in the Group 1 had a lower value of hemoglobin, platelet, and triglycerides, and a higher value of high-density lipoprotein cholesterol. Lesion and procedure information did not differ significantly between the 2 groups except for the rate of using mechanical support device. The comparison of medications in the present study is summarized in Table 2.

**Table 2. Medications.**

| Variables | All (n = 720) | Group 1 (n = 180) | Group 2 (n = 540) | p value |
|---|---|---|---|---|
| Medication at discharge | | | | |
| Aspirin | 575 (79.9%) | 139 (77.2%) | 436 (80.7%) | 0.33 |
| P2Y12 inhibitor | 679 (94.3%) | 171 (95.0%) | 508 (94.1%) | 0.71 |
| VKA | 105 (14.6%) | 35 (19.4%) | 70 (13.0%) | 0.04 |
| DOAC | 610 (84.7%) | 143 (79.4%) | 467 (86.5%) | 0.03 |
| ACE-i/ARB | 461 (64.0%) | 104 (57.8%) | 357 (66.1%) | 0.049 |
| β-blocker | 536 (74.6%) | 133 (73.9%) | 403 (74.8%) | 0.84 |
| Statins | 607 (84.3%) | 145 (80.6%) | 462 (85.6%) | 0.12 |
| Oral antidiabetic agents | 200 (27.8%) | 43 (23.9%) | 157 (29.1%) | 0.21 |
| Insulin | 62 (8.6%) | 16 (8.9%) | 46 (8.5%) | 0.88 |
| PPI | 629 (87.4%) | 158 (87.8%) | 471 (87.2%) | 0.90 |
| Steroid | 18 (2.5%) | 8 (4.4%) | 10 (1.9%) | 0.09 |
| Medication at 6 months | | | | |
| Aspirin | 318 (47.7%) | 72 (45.6%) | 246 (48.3%) | 0.58 |
| P2Y12 inhibitor | 438 (65.7%) | 93 (58.9%) | 345 (67.8%) | 0.04 |
| OAC | 649 (97.3%) | 152 (96.2%) | 497 (97.6%) | 0.40 |
| Duration of triple therapy | 78.9±101.1 | 69.3±98.6 | 82.1±101.8 | 0.14 |

ACE-i, angiotensin-converting enzyme inhibitor; ARB, angiotensin II receptor blocker; DOAC, direct oral anticoagulant; OAC, oral anticoagulant; PPI, proton pump inhibitor; VKA, vitamin K antagonist.

**Table 3. Adverse clinical events at 1 year.**

| Variables | All (n = 720) | Group 1 (n = 180) | Group 2 (n = 540) | OR* | 95% CI | p value† |
|---|---|---|---|---|---|---|
| NACE | 102 (14.2%) | 38 (21.1%) | 64 (11.9%) | 1.99 | 1.28–3.10 | 0.003 |
| MACE | 79 (11.0%) | 31 (17.2%) | 48 (8.9%) | 2.13 | 1.31–3.41 | 0.004 |
| All-cause death | 52 (7.2%) | 24 (13.3%) | 28 (5.2%) | 2.81 | 1.58–5.00 | <0.001 |
| Cardiovascular death | 27 (3.8%) | 12 (6.7%) | 15 (2.8%) | 2.50 | 1.15–5.45 | 0.02 |
| Myocardial infarction | 8 (1.1%) | 3 (1.7%) | 5 (0.9%) | 1.81 | 0.43–7.65 | 0.42 |
| Stent thrombosis | 5 (0.7%) | 1 (0.6%) | 4 (0.7%) | 0.75 | 0.08–6.74 | 1.00 |
| Ischemic stroke | 22 (3.1%) | 4 (2.2%) | 18 (3.3%) | 0.66 | 0.22–1.97 | 0.62 |
| Major bleeding (BARC 3 or 5) | 33 (4.6%) | 10 (5.6%) | 23 (4.3%) | 1.32 | 0.62–2.83 | 0.54 |
| All bleeding | 71 (9.9%) | 26 (14.4%) | 45 (8.3%) | 1.86 | 1.11–3.11 | 0.02 |

Values are expressed as n (%). BARC, Bleeding Academic Research Consortium; CI, confidence interval; MACE, major adverse cardiovascular events; NACE, net adverse clinical events; OR, odds ratio.

* The ratio of the event risk in the Group 1 to that in the Group 2

† Comparing the rate of each event between the Group 1and 2 with Fisher's exact test

The use of antihypertensive agents was more frequent along with the frequency of hypertension in the Group 2. The rate of DOAC rather than VKA was significantly lower in the Group 1 than in the Group 2. At 6 months follow-up, the percentage of patients taking P2Y12 inhibitors was significantly lower in the Group 1.

The crude incidence of clinical events is shown in Table 3.

The rates of NACE and MACE were significantly higher in the Group 1 (21.1% vs. 11.9%, p = 0.003, and 17.2% vs. 8.9%, p = 0.004). In the analysis of each component of MACE, all-cause death and the cardiovascular death more frequently occurred in the Group 1 than the counterpart (13.3% vs. 5.2%, p<0.001, and 6.7% vs. 2.8%, p = 0.002). While the incidence of all bleeding events was significantly higher in the Group 1 (14.4% vs. 8.3%, p = 0.02), that of major bleeding did not differ significantly between the 2 groups (5.6% vs. 4.3%, p = 0.54). Kaplan-Meier curve showed significantly higher cumulative rate of NACE in the Group 1 at one year after performing PCI (log-rank p = 0.002) (Fig 2). Whereas the cumulative rate of major bleeding was not significantly different between the 2 groups (log-rank p = 0.41), that of MACE was significantly higher in the Group 1 (log-rank p = 0.002) (S1 Fig). The multivariable Cox regression analysis revealed that male sex, low BMI value (<21.3 kg/m$^2$), current smoker, prior coronary artery bypass grafting, peripheral artery disease, severe CKD, thrombocytopenia, and moderate to severe anemia were independent predictors for NACE within 1 year after performing PCI (Table 4).

Although low BMI value independently promoted MACE (S1 Table), there was no significant association between low BMI value and major bleeding events (S2 Table). In sub-analyses which adjust several confounding variables, the rate of NACE was significantly higher in the patients with BMI value <21.3 kg/m$^2$ than those without in only men (22.0% vs. 11.8%, p = 0.006) (S3 Table). On the other hand, that was not significantly different in only women (18.8% vs. 12.0%, p = 0.31) (S3 Table). In the older adults (i.e. age ≥65 years), the patients with BMI value <21.3 kg/m$^2$ had significantly higher rate of NACE than their counterpart (21.0% vs. 12.1%, p = 0.007) (S4 Table). Baseline characteristics and information on medications after adjusting for age were summarized in S5 and S6 Tables. The rate of NACE was significantly higher after adjusting for age or excluding the patients with acute coronary syndrome or active cancer (S7 and S8 Tables). While the incidence of NACE did not differ significantly between the two groups of patients with low-dose prasugrel (14.8% vs. 13.7%, p = 0.85), there was a

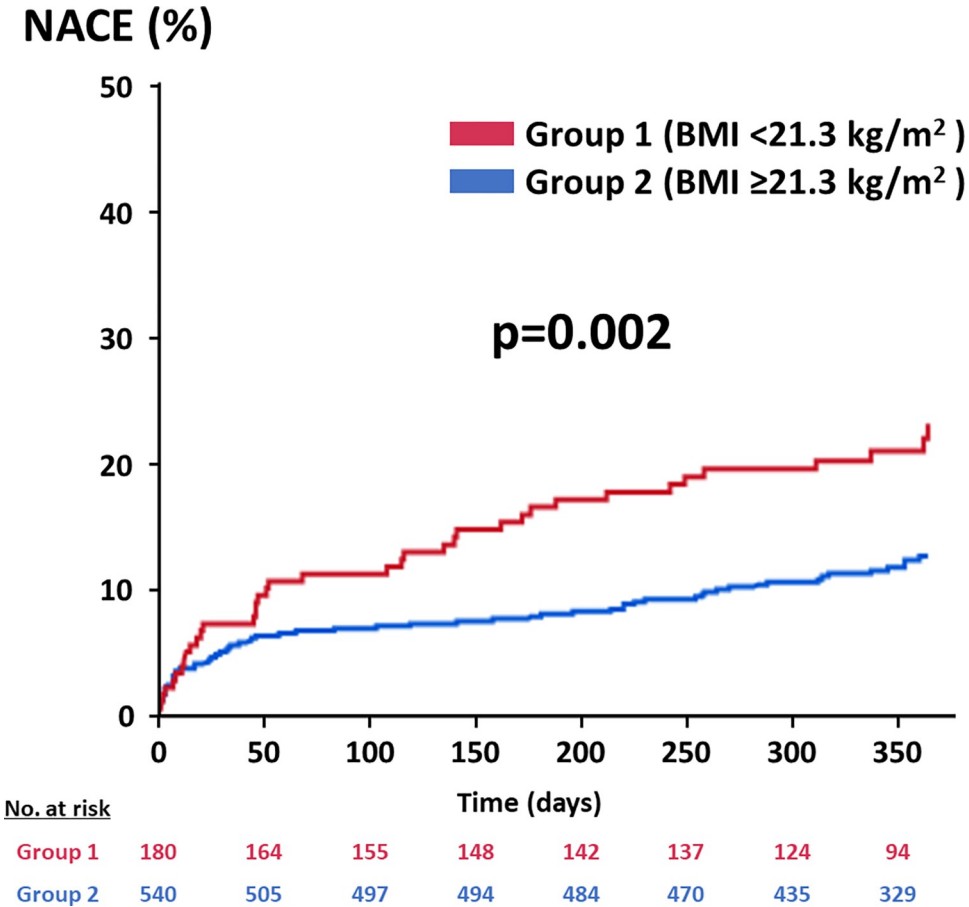

**Fig 2. Comparison of the cumulative incidence of the NACE between the Group 1 and 2.** BMI, body mass index; NACE, net adverse clinical events.

**Table 4. Multivariable Cox regression model of predictors for NACE.**

| Variables | Hazard ratio | 95% CI | p value |
|---|---|---|---|
| Age (per year) | 1.02 | 0.99–1.05 | 0.08 |
| Male sex | 1.34 | 0.79–2.30 | 0.29 |
| Low BMI ($< 21.3$ kg/m$^2$) | 1.66 | 1.08–2.57 | 0.02 |
| Diabetes | 1.20 | 0.79–1.83 | 0.39 |
| Current smoking | 1.77 | 1.05–2.98 | 0.03 |
| Prior CABG | 2.23 | 1.15–4.33 | 0.02 |
| Peripheral artery disease | 1.92 | 1.07–3.45 | 0.03 |
| Prior heart failure | 0.84 | 0.52–1.37 | 0.49 |
| Severe CKD | 1.97 | 1.10–3.52 | 0.02 |
| Thrombocytopenia | 6.72 | 3.04–14.9 | <0.001 |
| Moderate to severe Anemia | 2.02 | 1.24–3.30 | 0.005 |
| Acute coronary syndrome | 1.16 | 0.77–1.77 | 0.48 |
| VKA at discharge | 0.97 | 0.55–1.68 | 0.90 |

BMI, body mass index; CABG, coronary artery bypass grafting; CKD, chronic kidney disease; CI, confidence interval; NACE, net adverse clinical events; VKA vitamin K antagonist.

significant difference in the incidence of NACE between the two groups of the patients without low-dose prasugrel (26.3% vs. 10.0%, p<0.001) (S9 Table). In additional analyses which use other cut-off value of BMI, the rate of NACE was not significantly but numerically higher in the patients with BMI value <18.5 kg/m$^2$ than their counterpart (20.5% vs. 13.8%, p = 0.26) (S10 Table). When dividing the patients using median value of BMI in this study (i.e. 24.0 kg/m$^2$), NACE more frequently occurred in the patients with BMI value <24.0 kg/m$^2$ than those with BMI ≥24.0 kg/m$^2$ (16.9% vs. 11.4%, p = 0.04) (S10 Table).

## Discussion

In the present study, patients in the Group 1 had a higher risk of NACE and MACE, and BMI value <21.3 kg/m$^2$ was independent predictor for NACE and MACE within 1 year after performing PCI in patients under OAC therapy for AF. On the other hand, BMI value <21.3 kg/m$^2$ was not significantly associated with major bleeding events.

The relationship between BMI and clinical events has been validated in previous studies and a matter of debates. In traditional knowledge, patients with high BMI value, meaning obese patients, are likely to have an increased mortality and risk of cardiovascular events [1, 3]. However, several studies reported that obese patients had a favorable prognosis compared with patients with low or normal body weight in subjects with cardiovascular disease [4–7]. The Korean retrospective observational nationwide study, investigating the relationship between BMI value and clinical outcomes in Asian patients with AF receiving OAC, previously suggested that higher BMI value, per 5 kg/m$^2$ increase, was associated with lower risk of the composite clinical outcomes (hazard ratio, 0.751 [95% confidence interval, 0.706–0.799]) [4]. In contrast, patients having low BMI value, less than 18.5 kg/m$^2$, had a 1.4-fold risk of composite clinical outcomes compared with normal weight patients [4]. Another previous dose-response meta-analysis with almost 140,000 participants from 15 studies reported that, in cases undergoing PCI for coronary artery disease, obese patients had a lower risk of all-cause mortality than underweight patients [6]. These phenomena were observed in several studies and proposed as "obesity paradox". Although it has been considered that high usage rates of medications for comorbidities in obese patients, such as beta-blockers, angiotensin-converting enzyme inhibitors or angiotensin II receptor blockers, and statins, might influence the paradox [26, 29, 30], details were still unclear. In the present study, the usage rates of such medications were almost similar between the Group 1 and 2. On the other hand, patients in the Group 1 were older and had a higher rate of acute coronary syndrome, mechanical support, and higher CHA2DS2-VASc score than the counterpart, potentially resulting in the increased risk of NACE and MACE as similarly shown in previous reports [4–7]. In multivariable Cox regression analysis, low BMI value (i.e. <21.3 kg/m$^2$) was an independent predictor for NACE and MACE. Similar results were observed in a recent sub-analysis of a Korean prospective and randomized trial which compared the efficacy of antiplatelet drugs in the patients who underwent PCI for CAD, in which a combination risk of major cardiovascular and bleeding events was higher in the underweight patients and lower in obese patients compared with normal weight patients [26]. Considering these results, we believe that it is clinically relevant to recognize patients with low BMI value who undergo PCI with OAC therapy for AF as potentially high-risk subjects for adverse clinical events.

In many previous studies, low body weight and low BMI have been reported as independent risk factors for bleeding events in post-PCI or AF patients [4, 26, 31–33], and low body weight is proposed as one of the major criteria of high bleeding risk in recent Japanese guidelines on antithrombotic therapy in patients with CAD [16]. In the present study, however, the incidence of major bleeding events was not significantly higher in patients with BMI value <21.3

kg/m$^2$. The duration of triple therapy, using an oral anticoagulant and dual antiplatelet therapy simultaneously, was not significantly but numerically shorter in the Group 1 than the counterpart. In addition, the appropriately reduced dose of DOAC should have been used in patients with old age, low body weight, or kidney dysfunction, which were frequently observed in the Group 1, according to the guidelines [16]. These backgrounds might prevent bleeding events in the Group 1, resulting in the equivalent rate of major bleeding events between the 2 groups in the present study. Practically, a variety of associations between BMI and bleeding events have been previously reported. There are several reports that high BMI is associated with an increased risk of bleeding events [5, 34], that both low and high BMI can accelerate the bleeding risk (i.e. U-shaped association) [35, 36], and that there is no relationship between BMI and bleeding events [37]. Therefore, no definite conclusion has yet been reached regarding the relationship between BMI and bleeding events. Further studies are warranted to clarify this issue in AF patients after PCI.

## Limitations

The present study has several limitations. First, since this was an observational study based on the registry data, confounding factors might influence on the results. Second, this study did not use the classification of BMI recommended by World Health Organization (WHO) because of the limited number of underweight (BMI <18.5 kg/m$^2$) and obese (BMI ≥30.0 kg/m$^2$) patients: 44 patients (6.1%) had BMI <18.5 kg/m$^2$ and 69 (9.6%) had BMI ≥30.0 kg/m$^2$ in the present study [38]. When using the WHO standard cut-off, the rate of NACE was not significantly but numerically higher in the underweight patients than their counterpart (S10 Table). However, a previous individual-participant data meta-analysis which evaluated association of all-cause mortality with BMI in populations without chronic diseases demonstrated that the BMI value with the lowest risk of mortality gradually increased as age increased [39]. Similar results were shown in another study with Japanese population [40]. On this basis, the target BMI value for elderly people was higher than younger people as per the recommendation from the Health, Labor and Welfare Ministry in Japan [39, 40]. In that recommendation, 21.5 kg/m$^2$ is the proposed value for patients more than 65 years old, which applies to most of the patients in the present study, whose mean age was 74.0±9.0 years old. Given that the cut-off value defining the Group 1 in the present study (i.e. 21.3 kg/m$^2$) was in line with the target value for Japanese elderly people (i.e. 21.5 kg/m$^2$), it was thought that the classification of BMI in the present study was acceptable. Third, the small number of bleeding events did not allow inclusion of a sufficient number of variables in multivariable analysis which evaluated the factors associated with major bleeding events. Finally, because this study was a registry study, medical treatment was left to each physician. It was difficult to unify the characteristics of included patients, such as baseline characteristics, strategy of PCI, and medications. Although the choice and duration of antithrombotic therapy were decided based on thrombotic and bleeding events risk of each patient, the difference of antithrombotic therapy might be associated with the event rates seen in the present study.

## Conclusions

Among the patients undergoing PCI for CAD and requiring OAC for AF, low BMI value was independently associated with the incidence of NACE and MACE, although not with the incidence of major bleeding events. BMI value may be a useful indicator to predict major adverse clinical events.

## Supporting information

**S1 Table. Multivariable Cox regression model of predictors for MACE.**
(DOCX)

**S2 Table. Multivariable Cox regression model of predictors for major bleeding events.**
(DOCX)

**S3 Table. Adverse clinical events at 1 year in only men or women.**
(DOCX)

**S4 Table. Adverse clinical events at 1 year in the patients with <65 years or ≥65 years.**
(DOCX)

**S5 Table. Baseline characteristics after adjusted by age.**
(DOCX)

**S6 Table. Medications after adjusted by age.**
(DOCX)

**S7 Table. Adverse clinical events at 1 year after adjusted by age.**
(DOCX)

**S8 Table. Adverse clinical events at 1 year after excluding the patients with ACS or active cancer.**
(DOCX)

**S9 Table. Adverse clinical events at 1 year in the patients with and without low-dose prasugrel.**
(DOCX)

**S10 Table. Adverse clinical events at 1 year in the patients classified by WHO criteria or median value.**
(DOCX)

**S1 Fig. Comparison of the cumulative incidence of the MACE and major bleeding events between the Group 1 and 2.**
(TIF)

**S1 Dataset. All data used for analyses in this study.**
(XLSX)

## Acknowledgments

We thank Heidi N. Bonneau, RN, MS, CCA for her editorial review of the manuscript.

## Author Contributions

**Conceptualization:** Tatsuro Yamazaki, Hideki Kitahara, Yoshio Kobayashi.

**Data curation:** Tatsuro Yamazaki, Hideki Kitahara, Daichi Yamashita, Takanori Sato, Sakuramaru Suzuki, Takashi Hiraga, Tadahiro Matsumoto, Takahiro Kobayashi, Yuji Ohno, Junya Harada, Kenichi Fukushima, Tatsuhiko Asano, Naoki Ishio, Raita Uchiyama, Hirofumi Miyahara, Shinichi Okino, Masanori Sano, Nehiro Kuriyama, Masashi Yamamoto, Naoya Sakamoto, Junji Kanda, Yoshio Kobayashi.

**Formal analysis:** Tatsuro Yamazaki, Hideki Kitahara.

**Investigation:** Tatsuro Yamazaki, Hideki Kitahara, Daichi Yamashita, Takanori Sato, Sakura-maru Suzuki, Takashi Hiraga, Tadahiro Matsumoto, Takahiro Kobayashi, Yuji Ohno, Junya Harada, Kenichi Fukushima, Tatsuhiko Asano, Naoki Ishio, Raita Uchiyama, Hirofumi Miyahara, Shinichi Okino, Masanori Sano, Nehiro Kuriyama, Masashi Yamamoto, Naoya Sakamoto, Junji Kanda, Yoshio Kobayashi.

**Project administration:** Yoshio Kobayashi.

**Supervision:** Hideki Kitahara, Yoshio Kobayashi.

**Visualization:** Tatsuro Yamazaki.

**Writing – original draft:** Tatsuro Yamazaki.

**Writing – review & editing:** Hideki Kitahara, Yoshio Kobayashi.

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
