## [Decision Letter · Decision Letter 0]

21 Jun 2024

PONE-D-24-19204Relationship Between Body Mass Index and Clinical Events in Patients With Atrial Fibrillation Undergoing Percutaneous Coronary InterventionPLOS ONE

Dear Dr. Kitahara,

Thank you for submitting your manuscript to PLOS ONE. After careful consideration, we feel that it has merit but does not fully meet PLOS ONE’s publication criteria as it currently stands. Therefore, we invite you to submit a revised version of the manuscript that addresses the points raised during the review process.

We look forward to receiving your revised manuscript.

Kind regards,

Yoshihiro Fukumoto

Academic Editor

PLOS ONE

Reviewers' comments:

Reviewer's Responses to Questions

**Comments to the Author**

1. Is the manuscript technically sound, and do the data support the conclusions?

Reviewer #1: No

Reviewer #2: Yes

2. Has the statistical analysis been performed appropriately and rigorously? 

Reviewer #1: No

Reviewer #2: Yes

3. Have the authors made all data underlying the findings in their manuscript fully available?

Reviewer #1: No

Reviewer #2: Yes

4. Is the manuscript presented in an intelligible fashion and written in standard English?

Reviewer #1: Yes

Reviewer #2: Yes

5. Review Comments to the Author

Reviewer #1: In the present study, the authors evaluated the relationship between BMI and clinical events in patients who underwent PCI under OAC therapy for AF. Low BMI value was independently associated with the incidence of NACE and MACE, although not with the incidence of major bleeding events. The authors concluded that BMI value may be a useful indicator to predict major adverse clinical events in this specific patient population. It is certainly informative that low BMI may be relevant to cardiovascular risk, however, the novelty of this study is rather low and significant population biases may exist. The reviewer has comments as below.

# It is obvious that the Low BMI group has more severe clinical background. It includes higher age, more cancer patients, more anemic patients, more ACS, and more mechanical supports during PCI. Thus, the main result of the study that the Low BMI group had more clinical events, seems a matter of course. Do the results unchanged if they are age-matched or those with ACS and with cancer are excluded?

# Relevance of low BMI for NACE was significant in the multivariable Cox regression model in this study. What about for MACE?

# In table 3, sum of the number of MACE and major bleeding is not equal to NACE. Why?

# If the authors has a K-M analysis of MACE, it may be provided as a supplemental figure.

# The studied patients contains a special cohort with dual anti-thrombotic therapy with prasugrel. What is the number of them? No sub-analysis for them is provided. Why?

Reviewer #2: Review Comments

1.　Selection of BMI Cut-off Values

I understand from the authors' description in the Limitations section that the cutoff value of 21.3 for BMI is based on the first quartile of the study population. However, it is unclear to what extent this value can be generalized to other studies or clinical trials. Additionally, as it differs from the WHO standards, it makes international comparisons difficult. Therefore, to further validate the appropriateness of this cutoff value, it is recommended to conduct supplementary analyses using other cutoff values (e.g., 18.5 or 25) or analyses based on the second quartile.

2.　The need for further adjustment of confounding variables

In this study, multivariable Cox regression analysis was used to adjust for confounding factors, but further subgroup analyses are needed to more precisely evaluate the impact of sex and age. Since sex and age are likely to independently affect cardiovascular events and bleeding risks apart from BMI, additional analyses considering these variables are recommended. Specifically, please conduct subgroup analyses by sex to evaluate the relationship between BMI and clinical outcomes separately for men and women. Furthermore, instead of including age only as a continuous variable, it is advisable to perform age-stratified subgroup analyses to assess the relationship between BMI and clinical outcomes within each age group. This will provide a more accurate assessment of the relevance of BMI.

3.　Table3：Adverse clinical events at 1 year

In Table 3, the incidence rates of NACE and MACE are shown, but the confidence intervals (CI) for these rates are not provided. Adding 95% confidence intervals for the incidence rates of each event would make it easier to evaluate the precision of the results. Therefore, it is desirable to include these intervals.

4.　Regarding Group Classification and Naming

Generally, a BMI of 18.5 to 24.9 is defined as normal weight. From the context of the paper, it is understood that BMI 21.3 is used as the cutoff for grouping in this study. However, classifying the groups as Low BMI group and Normal-High BMI group feels somewhat inappropriate. It would be less confusing to categorize them as Group 1 and Group 2 or Group A and Group B, for example.

6. PLOS authors have the option to publish the peer review history of their article (what does this mean?). If published, this will include your full peer review and any attached files.

Reviewer #1: No

Reviewer #2: No

---

## [Author Response · Author response to Decision Letter 0]

4 Aug 2024

Responses to the comments of Reviewer #1

We thank the Reviewer for his/her time and input. We have modified the manuscript according to the suggestions. 

Reviewer#1 comment (1)

It is obvious that the Low BMI group has more severe clinical background. It includes higher age, more cancer patients, more anemic patients, more ACS, and more mechanical supports during PCI. Thus, the main result of the study that the Low BMI group had more clinical events, seems a matter of course. Do the results unchanged if they are age-matched or those with ACS and with cancer are excluded?

Author comment (1)

First and foremost, thank you so much for your time and commitment to improving the quality of our manuscript. As you pointed out, several clinical backgrounds in patients with low BMI value might be associated with poor clinical outcomes in the Group 1 (Low BMI group). According to the reviewer’s comments, we would like to provide the results of sub-analyses. First, propensity score matching was performed with propensity score calculated using logistic regression analysis to adjust between-group differences in age. Between the Group 1 (Low BMI group) and Group 2 (Normal-high BMI group), 177 patients were included in each group in the age-adjusted model. Baseline characteristics and information on medications were summarized in S5 and S6 Table. After age-matched, the rate of NACE and MACE was significantly higher in the Group 1 than their counterpart (S7 Table). Second, we provide the comparison of adverse events between 2 groups after excluding the patients with ACS and active cancer (S8 Table). While the rate of MACE did not differ significantly (12.8% vs. 6.9%, p=0.12), that of NACE was significantly higher in the Group 1 than the Group 2 (18.6% vs. 9.5%, p=0.04) (S8 Table). Considering these results, clinical outcomes did not change significantly after adjusting for age or excluding ACS and active cancer.

In the revised manuscript, we have added S5-S8 Table and the sentence “As a sub-analysis, propensity score matching was performed with propensity score calculated using logistic regression analysis to adjust between-group differences in age.” (page 10, paragraph 1, line 9-10) and modified the sentence “All statistical analyses were performed with JMP pro version 16.0 (SAS Institute Inc., Cary, USA) and EZR version 1.55 (Saitama Medical Center, Jichi Medical University, Saitama, Japan), which is a graphical user interface for R version 2.7-1 (The R Foundation for Statistical Computing, Vienna, Austria)” in the Methods (page 10, paragraph 1, line 14-17). In addition, we have added the sentence “Baseline characteristics and information on medications after adjusting for age were summarized in S5 and S6 Table. The rate of NACE was significantly higher after adjusting for age or excluding the patients with acute coronary syndrome or active cancer (S7 and S8 Table)” in the Results (page 16, paragraph 1, line 8-11).

Reviewer#1 comment (2)

Relevance of low BMI for NACE was significant in the multivariable Cox regression model in this study. What about for MACE?

Author comment (2)

In multivariable Cox regression analysis, low BMI value (<21.3kg/m2) was an independent predictor for MACE (HR 1.89, 95%CI [1.16-3.07], p=0.01). The results of the multivariable analysis for MACE and major bleeding are provided as S1 and S2 Table in the manuscript (page 16, paragraph 1, line 1-2). 

Reviewer#1 comment (3)

In table 3, sum of the number of MACE and major bleeding is not equal to NACE. Why?

Author comment (3)

I appreciate your meaningful comment. Since Table 3 describes the crude incidence of each event, it is possible that multiple events have occurred in the same case, which would result in the total number of MACE and major bleeding not being equal to NACE.

Reviewer#1 comment (4)

If the authors has a K-M analysis of MACE, it may be provided as a supplemental figure.

Author comment (4)

Thank you for your valuable comments. We have added the Kaplan-Meier curve for MACE and major bleeding as S1 Fig in the Results (page 15, paragraph 1, line 1-3), as you have indicated.

Reviewer#1 comment (5)

The studied patients contains a special cohort with dual anti-thrombotic therapy with prasugrel. What is the number of them? No sub-analysis for them is provided. Why?

Author comment (5)

In this study, treatment strategies, including details of medication, were left to each operator. In Japan, prasugrel has been used with OAC due to the approval of low-dose prasugrel (loading/maintenance dose: 20/3.75 mg/day), which is about one-third of that in other countries, as an antithrombotic therapy after performing PCI [Cardiovasc Interv Ther 2022; 37: 269-278. J Atheroscler Thromb 2015; 22: 557-569.]. Low-dose prasugrel was used in 352 (48.9%) patients at performing PCI. A comparison of the clinical outcomes between the Group 1 and Group 2 was conducted for each patient, with and without low-dose prasugrel and results were summarized in Table S9. While the incidence of NACE did not differ significantly between the two groups of patients with low-dose prasugrel (14.8% vs. 13.7%, p=0.85), there was a significant difference in the incidence of NACE between the two groups of patients without low-dose prasugrel (26.3% vs. 10.0%, p<0.001) (S9 Table). 

We added new references as reference No. 24 and 25. In addition, we have added S9 Table and the sentence “In Japan, prasugrel has been used with OAC due to the approval of low-dose prasugrel (loading/maintenance dose: 20/3.75 mg/day), which is about one-third of that in other countries, as an antithrombotic therapy after performing PCI [24,25].” in the Methods (page 8, paragraph 2, line 3 – page 9, paragraph 1, line 2) and “While the incidence of NACE did not differ significantly between the two groups of patients with low-dose prasugrel (14.8% vs. 13.7%, p=0.85), there was a significant difference in the incidence of NACE between the two groups of patients without low-dose prasugrel (26.3% vs. 10.0%, p<0.001) (S9 Table).”in the Results (page 16, paragraph 1, line 11 – page 17, paragraph 1, line 2).

Thank you again for your valuable contribution to elevating the quality of our manuscript.

Responses to the comments of Reviewer #2

We thank the Reviewer for his/her time and input. We have modified the manuscript according to the suggestions. 

Reviewer#2 comment (1)

Selection of BMI Cut-off Values

I understand from the authors' description in the Limitations section that the cutoff value of 21.3 for BMI is based on the first quartile of the study population. However, it is unclear to what extent this value can be generalized to other studies or clinical trials. Additionally, as it differs from the WHO standards, it makes international comparisons difficult. Therefore, to further validate the appropriateness of this cutoff value, it is recommended to conduct supplementary analyses using other cutoff values (e.g., 18.5 or 25) or analyses based on the second quartile.

Author comment (1)

First and foremost, thank you so much for your time and commitment to improving the quality of our manuscript. Following the reviewer's comments, we would like to provide the results of the sub-analyses. First, we compared clinical outcomes using the WHO standard cut-off. In this study population, 44 patients (6.1%) were defined as underweight patients (i.e. BMI <18.5 kg/m2) according to WHO criteria. The rate of NACE was not significantly but numerically higher in the patients with BMI value <18.5 kg/m2 than their counterpart (20.5% vs. 13.8%, p=0.26) (S10 Table). Given the common use of WHO criteria in the assessment of BMI, we agree with the reviewer that the results of the assessment using WHO criteria should be presented. Second, we used the value of second quartile to divide the patients into two groups. The second quartile value of BMI in this study population was 24.0 kg/m2 and the comparison of clinical outcomes between the patients with BMI value <24.0 kg/m2 and ≥24.0 kg/m2 is summarized in S10 Table. The rate of NACE was significantly higher in the patients with BMI value <24.0 kg/m2 than those with BMI ≥24.0 kg/m2 (S10 Table).

We have newly added S10 Table and added the sentence “In additional analyses which use other cut-off value of BMI, the rate of NACE was not significantly but numerically higher in the patients with BMI value <18.5 kg/m2 than their counterpart (20.5% vs. 13.8%, p=0.26) (S10 Table). When dividing the patients using median value of BMI in this study (i.e. 24.0 kg/m2), NACE more frequently occurred in the patients with BMI value <24.0 kg/m2 than those with BMI ≥24.0 kg/m2 (16.9% vs. 11.4%, p=0.04) (S10 Table).” in the Results (page 17, paragraph 1, line 2-7). In addition, the authors modified sentence “Second, this study did not use the classification of BMI recommended by World Health Organization (WHO) because of the limited number of underweight (BMI <18.5 kg/m2) and obese (BMI ≥30.0 kg/m2) patients: 44 patients (6.1%) had BMI <18.5 kg/m2 and 69 (9.6%) had BMI ≥30.0 kg/m2 in the present study [38]. When using the WHO standard cut-off, the rate of NACE was not significantly but numerically higher in the underweight patients than their counterpart (S10 Table).” in the Discussion (page 20, paragraph 2, line 2-8). 

Reviewer#2 comment (2)

The need for further adjustment of confounding variables

In this study, multivariable Cox regression analysis was used to adjust for confounding factors, but further subgroup analyses are needed to more precisely evaluate the impact of sex and age. Since sex and age are likely to independently affect cardiovascular events and bleeding risks apart from BMI, additional analyses considering these variables are recommended. Specifically, please conduct subgroup analyses by sex to evaluate the relationship between BMI and clinical outcomes separately for men and women. Furthermore, instead of including age only as a continuous variable, it is advisable to perform age-stratified subgroup analyses to assess the relationship between BMI and clinical outcomes within each age group. This will provide a more accurate assessment of the relevance of BMI.

Author comment (2)

Thank you for your comment. The author agrees with reviewer’s comment that additional analyses should be conducted, correcting for the effects of sex and age. First, the author conducted subgroup analysis to investigate the relationship between BMI value and clinical outcomes separately for men and women. In only men (n=580), 132 (22.8%) patients had low BMI value (i.e. <21.3 kg/m2) and the rate of NACE was significantly higher in the patients with BMI value <21.3 kg/m2 (22.0% vs. 11.8%, p=0.006) (S3 Table). On the other hand, in only women (n=140), low BMI value was observed in 48 (34.3%) patients and the rate of NACE was not significantly different between the patients with and without BMI value <21.3 kg/m2 (18.8% vs. 12.0%, p=0.31) (S3 Table). Although the multivariable analysis in the present study did not show sex as an independent predictor for NACE (Table 4), the results of sub-analyses might suggest that the impact of BMI on clinical outcomes differs between men and women. Second, according to reviewer’s comment, we performed age-stratified subgroup analyses to assess the relationship between BMI value and clinical outcomes. We divided the patients into 2 groups as follows: age <65 years group (n=90) and age ≥65 years group (n=630) because the cases over 65 years were defined as older adults in WHO criteria. In the patients with age <65 years, the rate of NACE did not differ significantly between the patients with and without BMI value <21.3 kg/m2 (23.1% vs. 10.4%, p=0.19) (Table S4). On the other hand, in the older adults (i.e. age ≥65 years), the rate of NACE was significantly higher in the Low BMI group than their counterpart (21.0% vs. 12.1%, p=0.007) (S4 Table), suggesting that BMI value may be associated with clinical outcome in the population limited to older people. These results provide new insights into the relationship between BMI value and clinical outcomes.

We have newly added S3 and S4 Table as supplemental materials. Additionally, we have added the sentence “In sub-analyses which adjust several confounding variables, the rate of NACE was significantly higher in the patients with BMI value <21.3 kg/m2 than those without in only men (22.0% vs. 11.8%, p=0.006) (Table S3). On the other hand, that was not significantly different in only women (18.8% vs. 12.0%, p=0.31) (S3 Table). In the older adults (i.e. age ≥65 years), the patients with BMI value <21.3 kg/m2 had significantly higher rate of NACE than their counterpart (21.0% vs. 12.1%, p=0.007) (S4 Table).” in the Results (page 16, paragraph 1, line 2-8)

Reviewer#2 comment (3)

Table3：Adverse clinical events at 1 year

In Table 3, the incidence rates of NACE and MACE are shown, but the confidence intervals (CI) for these rates are not provided. Adding 95% confidence intervals for the incidence rates of each event would make it easier to evaluate the precision of the results. Therefore, it is desirable to include these intervals.

Author comment (3)

I appreciate your meaningful comment. The authors agree with the reviewer’s suggestion that addition of 95% confidence intervals for the incidence rates of each event would be a better way of assessing the precision of the results. We conducted additional analyses to provide odds ratio and 95% confidence intervals in the comparison of the incidence rate of clinical events between the Group 1 (Low BMI group) and Group 2 (Normal-high BMI group) (Table 3_revised).

Reviewer#2 comment (4)

Regarding Group Classification and Naming

Generally, a BMI of 18.5 to 24.9 is defined as normal weight. From the context of the paper, it is understood that BMI 21.3 is used as the cutoff for grouping in this study. However, classifying the groups as Low BMI group and Normal-High BMI group feels somewhat inappropriate. It would be less confusing to categorize them as Group 1 and Group 2 or Group A and Group B, for example.

Author comment (4)

As you pointed out, the name of each group, “Low BMI group” and “Normal-High BMI”, may be inappropriate and cause confusing. We renamed “Low BMI group” to “Group 1” and “Normal-high BMI group” to “Group 2”. 

We have modified the sentence “Patients who had less than the first quartile value of BMI were classified as “Group 1” and remaining patients were classified as “Group 2” (Fig 1).” In the Methods (page 8, paragraph 1, line 1-3).

Thank you again for your valuable contribution to elevating the quality of our manuscript.

---

## [Decision Letter · Decision Letter 1]

19 Aug 2024

Relationship Between Body Mass Index and Clinical Events in Patients With Atrial Fibrillation Undergoing Percutaneous Coronary Intervention

PONE-D-24-19204R1

Dear Dr. Kitahara,

We’re pleased to inform you that your manuscript has been judged scientifically suitable for publication and will be formally accepted for publication once it meets all outstanding technical requirements.

Kind regards,

Yoshihiro Fukumoto

Academic Editor

PLOS ONE

Reviewers' comments:

Reviewer's Responses to Questions

**Comments to the Author**

1. If the authors have adequately addressed your comments raised in a previous round of review and you feel that this manuscript is now acceptable for publication, you may indicate that here to bypass the “Comments to the Author” section, enter your conflict of interest statement in the “Confidential to Editor” section, and submit your "Accept" recommendation.

Reviewer #1: All comments have been addressed

Reviewer #2: All comments have been addressed

2. Is the manuscript technically sound, and do the data support the conclusions?

Reviewer #1: Yes

Reviewer #2: Yes

3. Has the statistical analysis been performed appropriately and rigorously? 

Reviewer #1: Yes

Reviewer #2: Yes

4. Have the authors made all data underlying the findings in their manuscript fully available?

Reviewer #1: Yes

Reviewer #2: Yes

5. Is the manuscript presented in an intelligible fashion and written in standard English?

Reviewer #1: Yes

Reviewer #2: Yes

6. Review Comments to the Author

Reviewer #1: The authors well responded to the reviewers' comments and significantly improved their manuscript. I appreciate a bunch of works of the authors. The reviewer has no further comments.

Reviewer #2: Dear Authors

Thank you for your thorough and thoughtful responses to my comments. I appreciate the additional analyses you have conducted, as well as the clarifications you have made throughout the manuscript. Your efforts to address the points raised have strengthened the study, and the new data and supplementary tables have provided valuable insights. I find the revisions and responses satisfactory, and I believe the manuscript is now suitable for publication.

7. PLOS authors have the option to publish the peer review history of their article (what does this mean?). If published, this will include your full peer review and any attached files.

Reviewer #1: No

Reviewer #2: No

---

## [Editor Report · Acceptance letter]

11 Sep 2024

PONE-D-24-19204R1 

PLOS ONE

Dear Dr. Kitahara, 

I'm pleased to inform you that your manuscript has been deemed suitable for publication in PLOS ONE. Congratulations! Your manuscript is now being handed over to our production team.

Kind regards, 

on behalf of

Dr. Yoshihiro Fukumoto 

Academic Editor

PLOS ONE